# Noncaloric monosaccharides induce excessive sprouting angiogenesis in zebrafish via foxo1a-marcksl1a signal

Xiaoning Wang[1†], Jinxiang Zhao[1,2†], Jiehuan Xu[1†], Bowen Li[3], Xia Liu[1]*, Gangcai Xie[3]*, Xuchu Duan[1]*, Dong Liu[1]*

[1]Affiliated Hospital of Nantong University, Nantong Laboratory of Development and Diseases, School of Life Science; Co-innovation Center of Neuroregeneration, Nantong University, Nantong, China; [2]Suqian First Hospital, Suqian, China; [3]Medical School, Nantong University, Nantong, China

**Abstract** Artificially sweetened beverages containing noncaloric monosaccharides were suggested as healthier alternatives to sugar-sweetened beverages. Nevertheless, the potential detrimental effects of these noncaloric monosaccharides on blood vessel function remain inadequately understood. We have established a zebrafish model that exhibits significant excessive angiogenesis induced by high glucose, resembling the hyperangiogenic characteristics observed in proliferative diabetic retinopathy (PDR). Utilizing this model, we observed that glucose and noncaloric monosaccharides could induce excessive formation of blood vessels, especially intersegmental vessels (ISVs). The excessively branched vessels were observed to be formed by ectopic activation of quiescent endothelial cells (ECs) into tip cells. Single-cell transcriptomic sequencing analysis of the ECs in the embryos exposed to high glucose revealed an augmented ratio of capillary ECs, proliferating ECs, and a series of upregulated proangiogenic genes. Further analysis and experiments validated that reduced *foxo1a* mediated the excessive angiogenesis induced by monosaccharides via upregulating the expression of *marcksl1a*. This study has provided new evidence showing the negative effects of noncaloric monosaccharides on the vascular system and the underlying mechanisms.

*For correspondence:
liuxia_fd@ntu.edu.cn (XL);
gangcai@ntu.edu.cn (GX);
dxd2002sk@ntu.edu.cn (XD);
liudongtom@gmail.com (DL)

[†]These authors contributed equally to this work

## eLife assessment

This **valuable** study investigates the effect of noncaloric monosaccharides, sugar substitutes that are commonly used by diabetic patients, on angiogenesis in the zebrafish embryo. The authors show that noncaloric monosaccharides and glucose similarly induce excessive blood vessel formation due to the increased formation of tip cells by endothelial cells through the foxo1a-marcksl1a pathway. This **solid** study is of interest for the medical community in charge of the prevention and of the treatment of diabetes and other metabolic diseases.

## Introduction

Diabetes mellitus (DM) encompasses a group of chronic diseases characterized by elevated blood glucose levels. Among patients with DM, cardiovascular complications, especially the direct and indirect effects of hyperglycemia on the human vascular network, persist as the primary cause of morbidity and mortality (*de Matheus et al., 2013*). The harmful effects of hyperglycemia are closely associated with both microvascular and macrovascular complications, including retinopathy, nephropathy, neuropathy, atherosclerosis, ischemic heart disease, stroke, and peripheral artery disease (*Fox et al., 2004*; *Rask-Madsen and King, 2013*). Endothelial cell dysfunction is a systemic pathological state

**eLife digest** Consuming too much sugar can damage blood vessels and contribute to diseases like diabetes and heart disease. Artificial sweeteners have been suggested as a healthier alternative, and are now included in many products like sodas and baked goods.

However, some studies have suggested that people who consume large amounts of artificial sweeteners also have an increased risk of cardiovascular disease. Others suggest individuals may also experience spikes in blood sugar levels similar to those observed in people with diabetes. Yet few studies have examined how artificial sweeteners affect the network of vessels that transport blood and other substances around the body.

To investigate this question, Wang, Zhao, Xu, et al. studied zebrafish embryos which had been exposed to sugar and a type of artificial sweetener known as non-caloric monosaccharides. Various imaging tools revealed that high levels of sugar caused the embryos to produce more new blood vessels via a process called angiogenesis. This excessive growth of blood vessels has previously been linked to diabetic complications, including cardiovascular disease.

Wang, Zhao, Xu, et al. found that zebrafish embryos exposed to several different non-caloric monosaccharides developed similar blood vessel problems. All the sweeteners tested caused immature cells lining the blood vessels to develop into active tip cells that promote angiogenesis. This led to more new blood vessels forming that branch off already existing veins and arteries.

These findings suggest that artificial sweeteners may cause the same kind of damage to blood vessels as sugar. This may explain why people who consume a lot of artificial sweeteners are at risk of developing heart disease and high blood sugar levels. Future studies could help scientists learn more about how genetics or other factors affect the health impact of sugars and artificial sweeteners. This may lead to a greater understanding of the long-term health effects of artificially sweetened foods.

exhibiting disrupted integrity, adhesion, altered proliferation capacity, migration, tube formation, and more (*Flammer et al., 2012*; *Kolluru et al., 2012*). High blood glucose levels over long periods have been demonstrated to be associated with vascular dysfunction both in vivo and in vitro (*Tesfamariam et al., 1990*; *Ting et al., 1996*).

The epidemiological evidence has indicated the positive correlation between risks of cardiovascular disease and DM with the consumption of sugar-sweetened beverages (SSBs) and 100% fruit juices, thereby emphasizing the concerns for the adverse effects of sugar intake on cardiometabolic risk factors, regardless of whether the sugar is added or naturally occurring (*Imamura et al., 2015*; *Malik, 2017*; *Narain et al., 2016*; *Larsson et al., 2014*; *Fung et al., 2009*). Artificially sweetened beverages (ASBs), which incorporate noncaloric sweeteners or low-caloric additives, have been suggested as healthy alternatives to SSBs (*Fakhouri et al., 2012*). ASBs contain sugar alcohols and polyols, such as sorbitol, xylitol, maltitol, mannitol, erythritol, isomalt, and lactitol. The consumption of ASBs worldwide has gradually increased in recent years (*Fakhouri et al., 2012*; *Sylvetsky et al., 2012*; *Moriconi et al., 2020*). However, accumulating studies in the last decade suggested that ASB consumption might be associated with an increased risk of cardiovascular events and diabetes (*Fung et al., 2009*; *de Koning et al., 2012*; *Fagherazzi et al., 2013*; *Gardener et al., 2012*; *Drouin-Chartier et al., 2019*; *de Koning et al., 2011*; *Hirahatake et al., 2019*; *Mossavar-Rahmani et al., 2019*; *Vyas et al., 2015*). Nevertheless, the underlying mechanisms responsible for these findings remain insufficiently documented.

The zebrafish has been recognized as a valuable animal model for studying metabolic diseases, such as hyperglycemia and diabetic complications, due to its functional conservation in glycol metabolism, pancreas structure, glucose homeostasis, adipose biology, and genetic similarities to mammals (*Zang et al., 2018*; *Barros et al., 2008*; *Elo et al., 2007*). The combination of embryonic transparency and transgenic lines, wherein endothelial cells are labeled specifically with fluorescent proteins, facilitates the high-resolution imaging analysis of vascular formation in vivo. Immersion of zebrafish in glucose solution has been found to induce diabetic complications, including vascular dysfunction (*Gleeson et al., 2007*; *Alvarez et al., 2010*; *Jörgens et al., 2012*; *Jung et al., 2016*). Several recent studies have investigated the effects of high glucose on vascular function in the zebrafish model (*Jung et al., 2016*; *Heckler and Kroll, 2017*; *Jörgens et al., 2015*). However, the association between

noncaloric monosaccharides and vascular dysfunctions, such as excessive angiogenesis, has not been elucidated. Here, we have successfully established a short-term zebrafish model that exhibits significantly excessive angiogenesis similar to the phenotypes observed in proliferative diabetic retinopathy (PDR) induced by glucose treatment. Using this model, we examined the effects of noncaloric monosaccharides on blood vessel development and investigated the molecular mechanisms. Our results provided new evidence for the negative roles of caloric and noncaloric monosaccharides on vascular development.

## Results

### Establishment of a short-term model of high glucose-induced excessive angiogenesis

To establish the short-term zebrafish hyperangiogenenic model induced by high glucose treatment, we immersed the *Tg(fli1aEP:EGFP-CAAX)^{ntu666}* embryos, a transgenic line wherein the endothelial cells were labeled with membrane-bound GFP (*Figure 1—figure supplement 1*), in glucose solution within a wide range of concentrations and time windows (*Figure 1—figure supplement 2*). We subsequently measured the glucose concentration in the embryos. We found that the glucose concentration in the embryos treated with high glucose was significantly higher than that in the control group (*Figure 1—figure supplement 3*). We observed that exposing zebrafish embryos at either 24 hr post fertilization (hpf) or 48 hpf to a 6% D-glucose treatment for a duration exceeding 48 hr led to dramatically increased formation of blood vessels (*Figure 1*, *Videos 1 and 2*), especially intersegmental vessels (ISVs) in the indicated area (*Figure 1b*). The hyperbranched endothelial cells were observed to sprout from existing vessels, including the ISVs, dorsal aorta (DA), and dorsal lateral anastomotic vessel (DLAV) (*Figure 1*) in embryos treated with high glucose.

Additionally, these ectopically branched angiogenic sprouts were not perfused by blood flow. Despite the abnormal vessel formation, no significant developmental defects were observed in these treated embryos when examined under a bright-field microscope (*Figure 1—figure supplement 4a–c*). Moreover, no excessive angiogenic phenotype was observed in the embryos treated with 1%, 2%, 3%, and 4% D-glucose within the corresponding time frame (*Figure 1—figure supplement 5*).

### Fructose and noncaloric monosaccharides induce excessive angiogenesis

Fructose is a ketonic monosaccharide that is an energy source for living organisms. Therefore, our study investigated the potential effects of fructose on vascular dysfunction in comparison to glucose. The result demonstrated fructose-induced excessive angiogenesis in zebrafish embryos (*Figure 2—figure supplement 1*). Wondering whether the effects of glucose and fructose on vascular development were mediated by metabolic events, we then conducted the same tests by using other noncaloric monosaccharides, including L-glucose, D-mannose, D-ribose, and L-arabinose, which animals could not digest. Interestingly, we observed that all these noncaloric monosaccharides could induce excessive angiogenesis, among which the L-glucose purchased from two companies resulted in a similar phenotype as efficiently as D-glucose did (*Figure 2a–h*). To rule out the effect of osmotic pressure, we treated zebrafish embryos with isotonic disaccharides, including lactose, maltose, and sucrose, which did not cause a significant excessive angiogenic phenotype (*Figure 1—figure supplement 4d and h*; *Figure 2—figure supplement 2*). However, higher concentration disaccharide treatment can also cause excessive angiogenesis in zebrafish embryos (*Figure 2—figure supplement 3*). In addition, we also tested the effects of pyruvic acid but did not observe the excessive angiogenic phenotype in the embryos treated with pyruvic acid solution at 50 nM to 50 µM concentration (*Figure 2—figure supplement 4*). Furthermore, we examined the arterial and venous identity of the hyperbranched vessels via live imaging analysis of the high glucose-treated *Tg(flt1:YFP::kdrl:ras-mCherry)* line, in which the YFP expression in the artery was relatively higher than that in the vein (*Krueger et al., 2011*). The result revealed that the hyperbranched ectopic vessels comprised arteries and veins (*Figure 2i and j*).

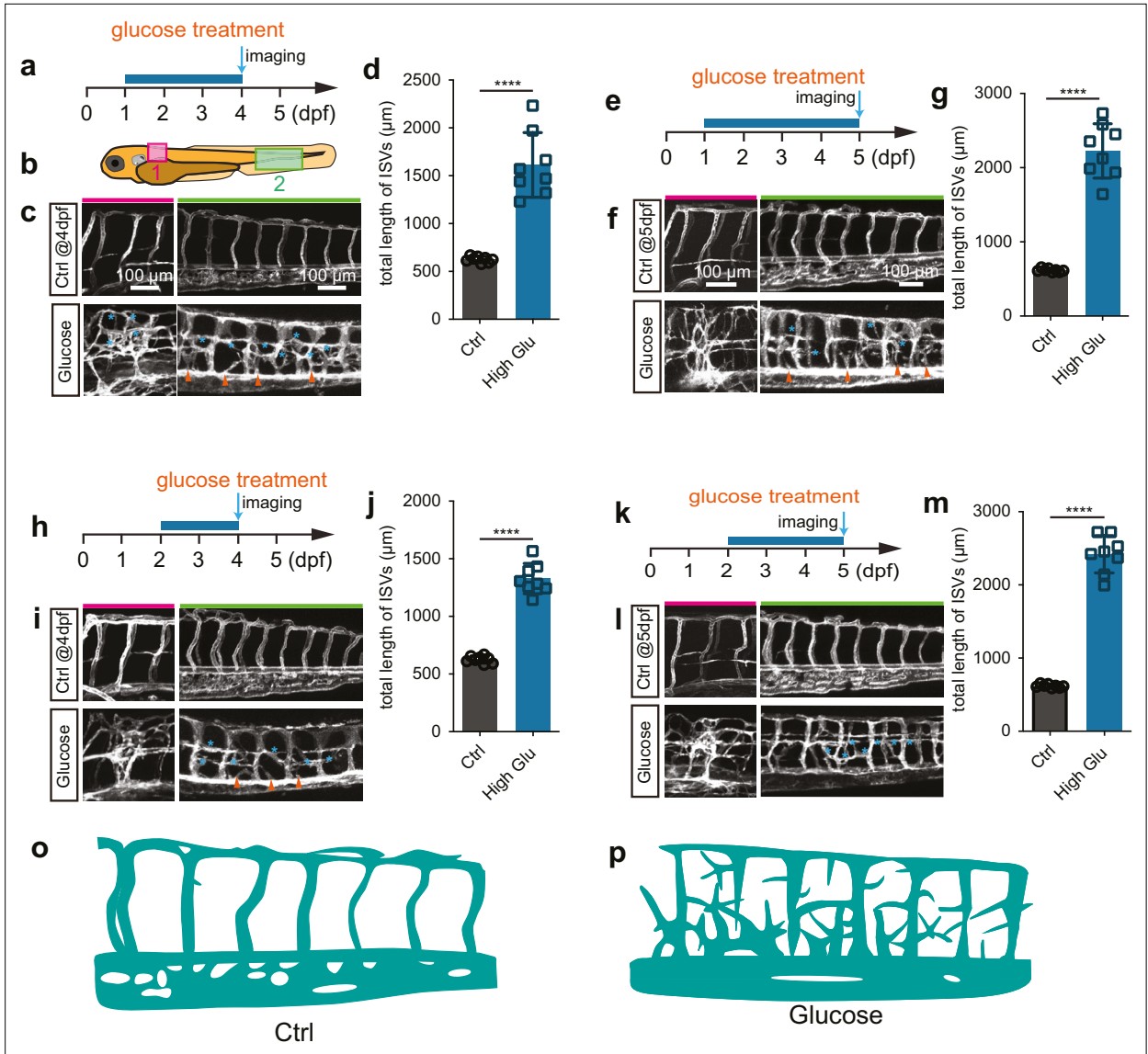

**Figure 1.** Glucose treatment caused excessive angiogenesis in zebrafish. (**a**) A diagram showing the glucose treatment time window and imaging time point. (**b**) A diagram indicating the imaging positions of the zebrafish embryos. (**c**) Confocal imaging analysis of the control and glucose-treated embryos. The red bar indicates position 1; the green bar indicates position 2. Arrowheads indicate the ectopic branching from the dorsal aorta. Stars indicate the ectopic vessels from intersegmental vessels (ISVs) and dorsal lateral anastomotic vessels (DLAVs). (**d**) Statistical analysis of the total length of ISVs in control and glucose-treated embryos (n=8). t-test, ****p<0.0001. (**e**) A diagram showing the glucose treatment time window and imaging time point. (**f**) Confocal imaging analysis of the control and glucose-treated embryos. The red bar indicates position 1; the green bar indicates position 2. Arrowheads indicate the ectopic branching from the dorsal aorta. Stars indicate the ectopic vessels from ISVs and DLAVs. (**g**) Statistical analysis of the total length of ISVs in control and glucose-treated embryos (n=8). t-test, ****p<0.0001. (**h**) A diagram showing the glucose treatment time window and imaging time point. (**i**) Confocal imaging analysis of the control and glucose-treated embryos. The red bar indicates position 1; the green bar indicates position 2. Arrowheads indicate the ectopic branching from the dorsal aorta. Stars indicate the ectopic vessels from ISVs and DLAVs. (**j**) Statistical analysis of the total length of ISVs in control and glucose-treated embryos (n=8). t-test, ****p<0.0001. (**k**) A diagram showing the glucose treatment time window and imaging time point. (**l**) Confocal imaging analysis of the control and glucose-treated embryos. The red bar indicates position 1; the green bar indicates position 2. Arrowheads indicate the ectopic branching from the dorsal aorta. Stars indicate the ectopic vessels from ISVs and DLAVs. (**m**) Statistical analysis of the total length of ISVs in control and glucose-treated embryos (n=8). t-test, ****p<0.0001. (**o**) A diagram showing the blood vessels in position 2 indicated in panel b of control embryos. (**p**) A diagram showing the blood vessels in position 2 indicated in panel b of high glucose-treated embryos.

The online version of this article includes the following figure supplement(s) for figure 1:

**Figure supplement 1.** Confocal imaging analysis of *Tg(fli1aEP:EGFP-CAAX)*[ntu666] embryos at 48 hr post fertilization (hpf) and 72 hpf.

*Figure 1 continued on next page*

## High glucose promotes quiescent endothelial differentiation into tip cells

Given that a high glucose shock has been observed to induce excessive angiogenesis in 48 hpf embryos, it was hypothesized that the shock might play a crucial role in regulating the differentiation of quiescent endothelial cells (ECs) into active tip cell-like cells and their subsequent behaviors. To investigate whether this was the case, we observed the behaviors of these ECs by confocal time-lapse imaging analysis. As shown in the result, in control *Tg(fli1aEP:EGFP-CAAX)*[ntu666] embryos, no significant activation of tip cells in the angiogenic sprouts was observed in the generated ISVs, DA, and DLAV in the embryos aged from 48 hpf to 5 dpf. Moreover, only a few ECs in established ISVs, DA, and DLAV extended filopodia, which quickly retracted (*Figure 3a–c*, *Video 3*). However, many ECs initiated sprouting angiogenesis in the high glucose-treated embryos, extended dynamic filopodia to sense the surroundings, and formed excessive ectopic blood vessels (*Figure 3*, *Video 4*). In a snapshot, we observed that some of the ECs protruded long and intricate sprouts simultaneously (*Figure 3f*), and nearly all the ECs within an ISV underwent the outgrowth of filopodia in some extreme cases (*Figure 3g*), suggesting that the high glucose treatment induced the endothelial differentiation into tip cell-like cells. Furthermore, we observed that these outgrowths of the ectopic angiogenic sprouts could establish a connection to the neighboring sprouts and vessels and thereby form complicated vascular structures (*Figure 1c, f, i, l, and p*).

## Single-cell transcriptomic sequencing analysis of the ECs isolated from glucose-treated embryos

We did a single-cell transcriptomic sequencing analysis to gain more insight into the potential mechanism through which glucose activates the ECs. Due to the limited presence of ECs within the zebrafish embryos, the analysis of these cells poses a challenge. First, we isolated the EGFP-positive cells from control and high glucose-treated embryos. Following the proteolytic dissociation of embryos, the EGFP-positive cells were isolated by fluorescence-activated cell sorting. Around 300–500 zebrafish embryos were used for the ECs collection for each stage. The isolated ECs were analyzed using a large-scale scRNA-seq (10X Genomics) platform, and the pipeline is illustrated in the diagram (*Figure 4a*).

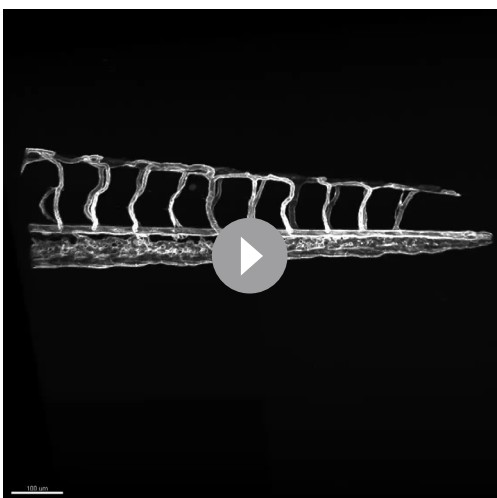

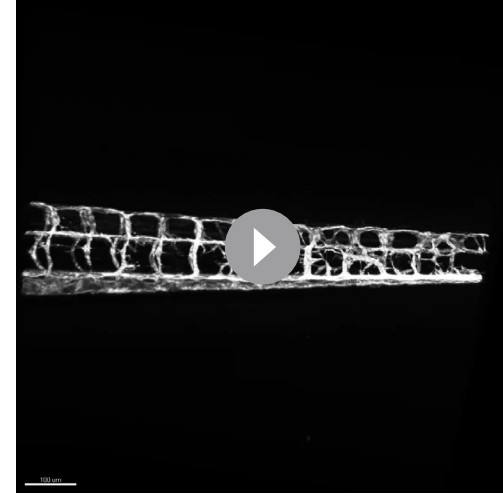

**Video 1.** The 3D structure of the blood vessels in the control *Tg(fli1aEP:EGFP-CAAX)*[ntu666] embryos.
https://elifesciences.org/articles/95427/figures#video1

**Video 2.** The 3D structure of the blood vessels in the glucose-treated *Tg(fli1aEP:EGFP-CAAX)*[ntu666] embryos.
https://elifesciences.org/articles/95427/figures#video2

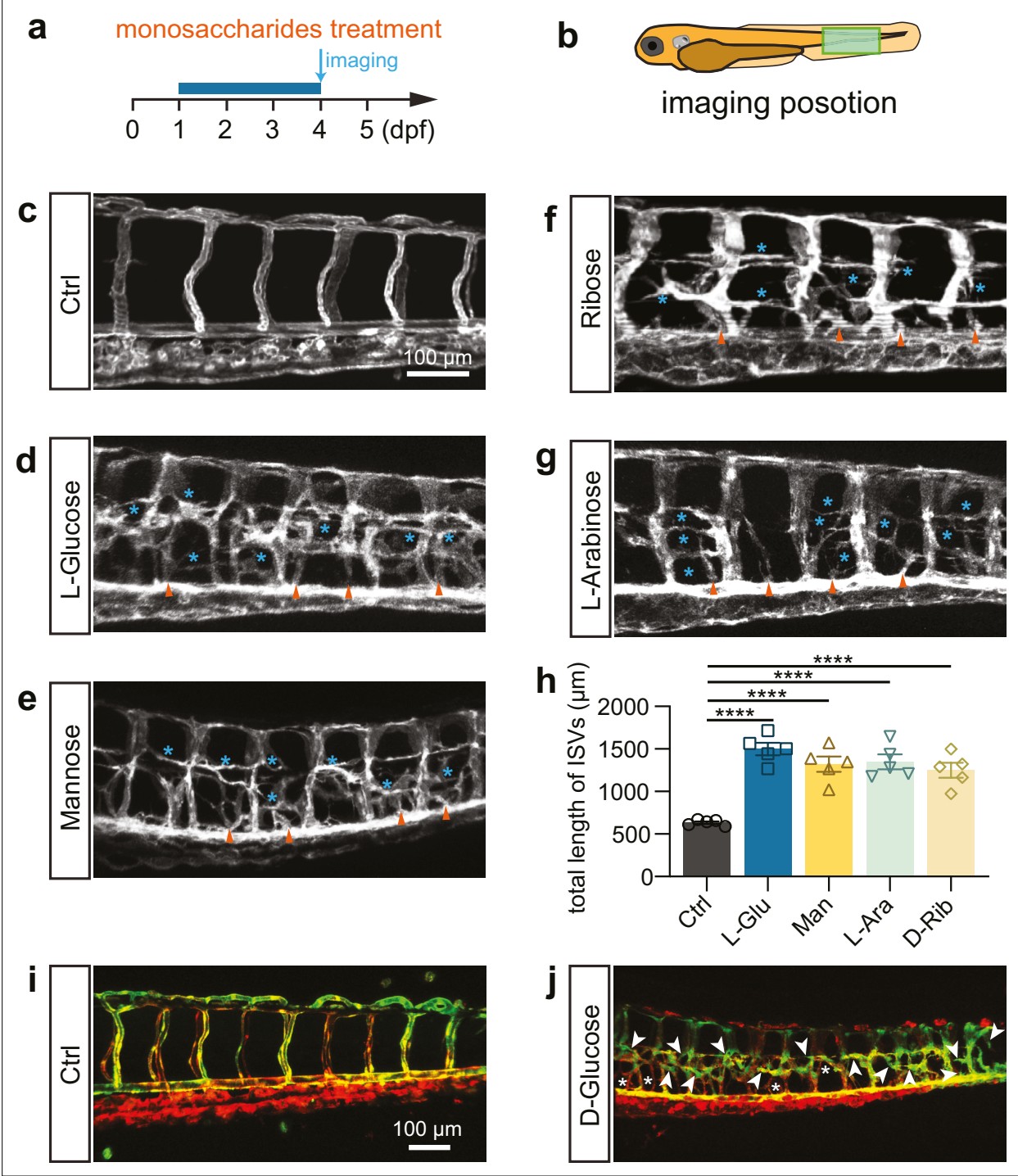

**Figure 2.** L-Glucose and mannose treatment caused excessive angiogenesis as well. (**a**) A diagram showing the monosaccharides treatment time window and imaging time point. (**b**) A diagram indicating the imaging position of the zebrafish embryos. (**c–g**) Confocal imaging analysis of the control and monosaccharides, including L-glucose, D-mannose, D-ribose, and L-arabinose, treated embryos. Arrowheads indicate the ectopic branching from the dorsal aorta. Stars indicate the ectopic vessels from intersegmental vessels (ISVs). (**h**) Statistical analysis of the total length of ISVs in control and monosaccharides-treated embryos (n=5). (**i, j**) Confocal imaging analysis of the control and glucose-treated embryos. Arrowheads indicate the ectopic branching of arteries. Stars indicate the ectopic branching of veins. t-test, ****p<0.0001.

The online version of this article includes the following figure supplement(s) for figure 2:

**Figure supplement 1.** Fructose treatment caused excessive angiogenesis in zebrafish.

**Figure supplement 2.** Lactose and maltose treatment did not cause excessive angiogenesis in zebrafish.

*Figure 2 continued on next page*

*Figure 2 continued*

**Figure supplement 3.** Higher concentration sucrase and maltose treatment cause excessive angiogenesis in zebrafish embryos.

**Figure supplement 4.** Pyruvic acid treatment did not cause excessive angiogenesis in zebrafish.

Multiple criteria were applied to select the single cells, including the retention of the genes that were expressed (Unique Molecular Identifiers or UMI larger than 0) in at least three individual cells, the selection of cells with the gene expression count falling within the range of 500–3000, and the imposition of a threshold wherein the proportion of sequencing reads derived from the mitochondrial genome was limited to less than 5% (*Figure 4—figure supplement 1*, *Figure 4—figure supplement 2*, *Supplementary file 1*). Ultimately, 6006 ECs were selected for further analysis (*Supplementary file 2*).

Through clustering analysis of gene expression, these ECs were categorized into six clusters using UMAP. These clusters include cluster 0, which consists of arterial and capillary ECs; cluster1, comprising endocardium; cluster2, consisting of venous and lymphatic ECs; cluster3, comprising arch ECs; cluster4, encompassing proliferating ECs; and cluster5, consisting of vesicle-enriched ECs (*Figure 4b*). The endothelial marker gene *cdh5* was expressed in all the clusters (*Figure 4c*). The notch ligand *dlc* was highly expressed in arterial, capillary ECs, and arch ECs (*Figure 4d*). The *dab2* and *prox1* were mainly enriched in venous and lymphatic ECs (*Figure 4e and f*). The *cdk1*, a key player in cell cycle regulation, was specifically expressed in proliferating ECs (*Figure 4h*). It was revealed that the ratio of arterial and capillary ECs and proliferating ECs was increased in the high glucose-treated embryos (*Figure 4i and j*), consistent with the observation that glucose treatment resulted in excessive sprouting angiogenesis of ISVs. In addition, we examined tip cell marker genes in arterial and capillary ECs. The results showed that the expression of *esm1*, *cxcr4a*, and *apln* was significantly upregulated after high glucose treatment (*Figure 3h–k*), consistent with our observation that high glucose treatment induced the endothelial differentiation into tip cell-like cells (*Figure 3f and g*).

We also performed the whole embryo transcriptome sequencing after high D-glucose and L-glucose treatment. We analyzed and compared the differentially expressed genes (DEGs) of control, high D-glucose-treated, and high L-glucose-treated embryos. The results revealed that 1259 and 1074 genes were upregulated significantly in high D-glucose- and high L-glucose-treated embryos, respectively, compared with the control (*Figure 4—figure supplement 3*). After that, we analyzed the expression of the genes related to metabolic pathways and found significant alteration in the expression of several genes involved in gluconeogenesis, glycolysis, and oxidative phosphorylation (*Figure 4—figure supplement 4*).

## Foxo1a was significantly downregulated in arterial and capillary ECs

To identify the potential molecules responsible for increasing the proportion of arterial and capillary ECs in the embryos treated with glucose, we analyzed and compared the DEGs in arterial and capillary ECs of control and glucose-treated ECs. The results revealed that 1201 genes were upregulated and 523 genes were downregulated significantly (*Figure 5a*). Gene ontology (GO) analysis revealed that these DEGs were enriched in several biological processes, including regulation of actin filament organization, blood vessel morphogenesis, development, angiogenesis, etc. (*Figure 5b*).

Subsequently, we searched for transcription factors among the genes involved in the aforementioned biological processes that might participate in inducing excessive angiogenesis. It has been reported that the loss of function of *foxo1a* led to excessive angiogenesis (*Wilhelm et al., 2016*; *Rudnicki et al., 2018*). Our study also revealed that *foxo1a* was significantly downregulated in arterial and capillary ECs after high glucose treatment compared to the ECs marker gene *pecam1* (*Figure 5c–e*). The in situ hybridization (ISH) experiments further confirmed the decrease in *foxo1a* expression following treatment with high D-glucose and L-glucose (*Figure 5f*). To verify whether the downregulation of Foxo1a led to excessive angiogenesis in zebrafish embryos, we performed loss-of-function experiments targeting *foxo1a*. AS1842856, a cell-permeable inhibitor reported to block *FOXO1* transcription activity (*Zhao et al., 2019*), was administered to zebrafish embryos at 48 hpf, and the imaging was performed at 72 hpf. The results revealed significantly excessive angiogenesis in AS1842856-treated embryos compared with the control group, consistent with the results obtained from *foxo1a* MO injection (*Figure 5g–i*).

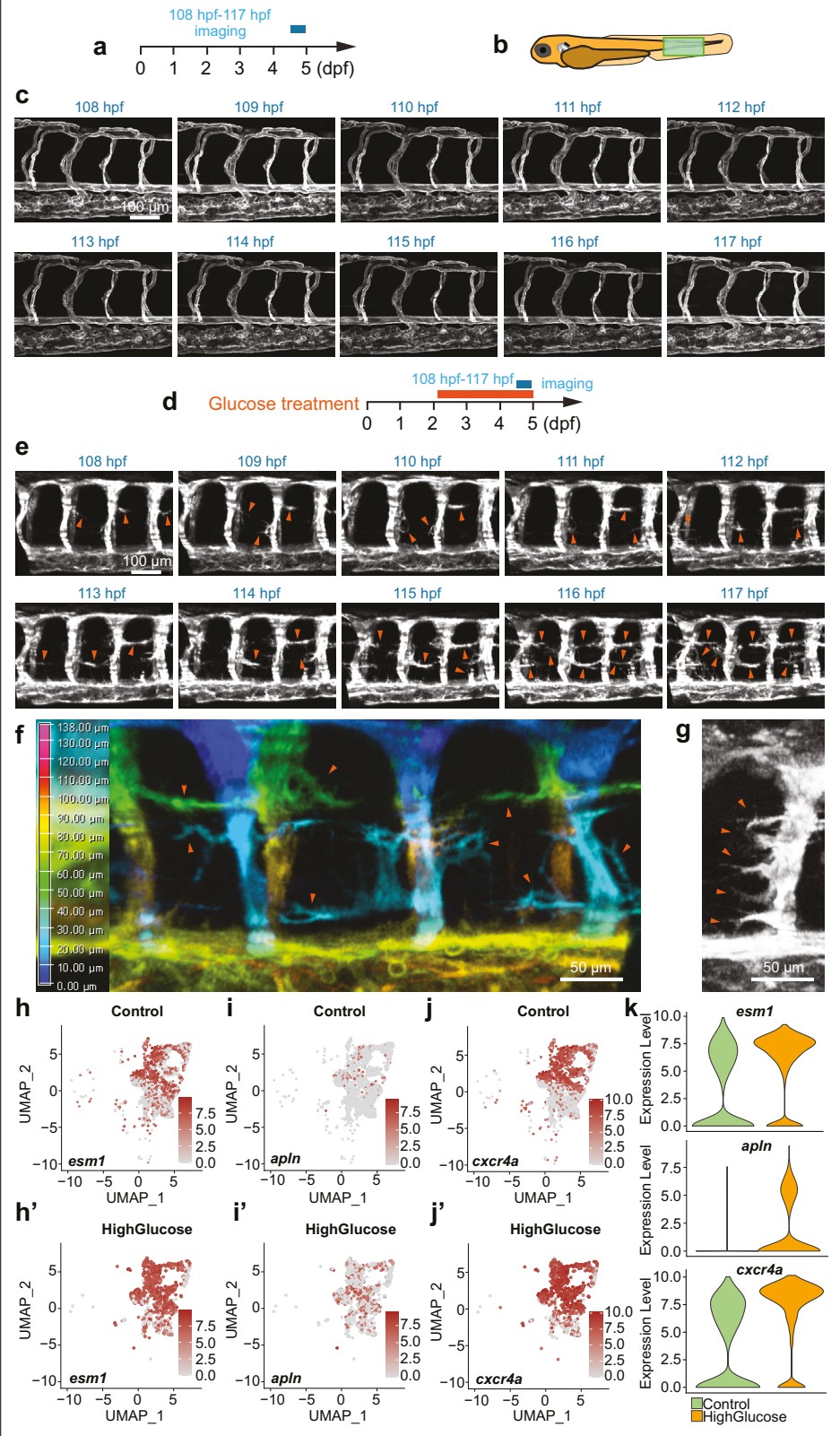

**Figure 3.** High glucose treatment induced endothelial differentiation into tip cell-like cells. (**a**) A diagram showing the confocal time-lapse imaging time window. (**b**) A diagram indicating the imaging position of the zebrafish embryos. (**c**) Confocal time-lapse imaging analysis of blood vessels in control *Tg(fli1aEP:EGFP-CAAX)^ntu666* embryos. (**d**) A diagram showing the glucose treatment time window and confocal time-lapse imaging time window. (**e**)

*Figure 3 continued on next page*

*Figure 3 continued*

Confocal time-lapse imaging analysis of blood vessels in glucose-treated *Tg(fli1aEP:EGFP-CAAX)*[ntu666] embryos. Arrowheads indicate the ectopic angiogenic branches. (**f**) A snapshot of confocal time-lapse imaging analysis of blood vessels in glucose-treated *Tg(fli1aEP:EGFP-CAAX)*[ntu666] embryos. Z stacks were used to make 3D color projections, where blue represents the most proximal (closest to the viewer), and red represents the most distal (farthest from the viewer). Arrowheads indicate ectopic angiogenic sprouts. (**g**) A snapshot of confocal time-lapse imaging analysis of an intersegmental vessel (ISV) in glucose-treated *Tg(fli1aEP:EGFP-CAAX)*[ntu666] embryos. Arrowheads indicate ectopic angiogenic sprouts. (**h-j'**) The feature plot of tip cell marker genes *esm1*, *apln* and *cxcr4a* of control and high glucose group in arterial and capillary ECs. (**k**) The violin plot of tip cell marker genes *esm1*, *apln* and *cxcr4a* of control and high glucose group in arterial and capillary ECs.

To further validate that the excessive angiogenesis induced by high glucose was attributed to Foxo1a deficiency, we performed rescue experiments. In detail, *foxo1a* was overexpressed in either whole embryos or ECs, driven by *hsp70l* and *fli1EP* promoter, respectively. We injected the overexpression construct into one-cell stage embryos followed by heat shock at 24 hpf and 48 hpf. The embryos were then treated with high glucose from 48 hpf to 96 hpf (*Figure 6a*). The results indicated that the gain of function of foxo1a in either whole embryos or ECs significantly and partially mitigated the excessive angiogenesis induced by high glucose treatment (*Figure 6b–j*).

## Monosaccharides induced excessive angiogenesis through the *foxo1a-marcksl1a* pathway

A previous study has reported that *marcksl1a* overexpression in ECs in zebrafish led to a significant increase in filopodia formation, similar to the phenotype we observed in response to high glucose treatment (*Kondrychyn et al., 2020*). Our analysis of the single-cell sequencing data revealed a significant upregulation of *marcksl1a* in arterial and capillary ECs following high glucose treatment, compared to the ECs marker gene *kdrl* (*Figure 7a and b*). The real-time quantitative PCR (qPCR) and ISH experiments further confirmed the elevated expression levels of *marcksl1a* following high D-glucose and L-glucose treatment (*Figure 7c and d*). Then, by constructing the transgenic zebrafish line *hsp70l:marcksl1a-p2A-mCherry::Tg(fli1a:EGFP-CAAX)* [ntu666], we conducted the experiments to overexpress *marcksl1a* in zebrafish and subsequently observed the vascular developmental phenotype. After 1 hr of heat shock at 24 hpf and confocal imaging analysis at 72 hpf, significantly increased blood vessel formation was observed in embryos overexpressing *marcksl1a*, compared with the control group (*Figure 7e–g*).

Given the results obtained from *marcksl1a* overexpression and loss of function of *foxo1a*, we hypothesized that *marcksl1a* might be a target

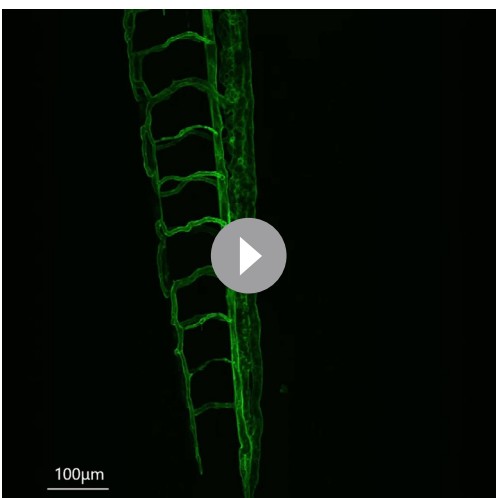

**Video 3.** Time-lapse imaging analysis of filopodia in the control *Tg(fli1aEP:EGFP-CAAX)*[ntu666] embryos. https://elifesciences.org/articles/95427/figures#video3

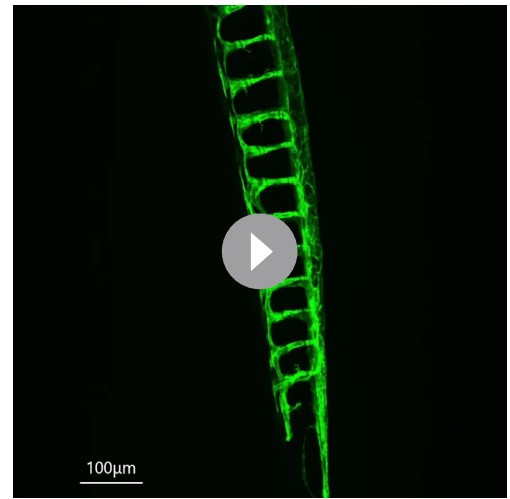

**Video 4.** Time-lapse imaging analysis of filopodia in the glucose-treated *Tg(fli1aEP:EGFP-CAAX)*[ntu666] embryos. https://elifesciences.org/articles/95427/figures#video4

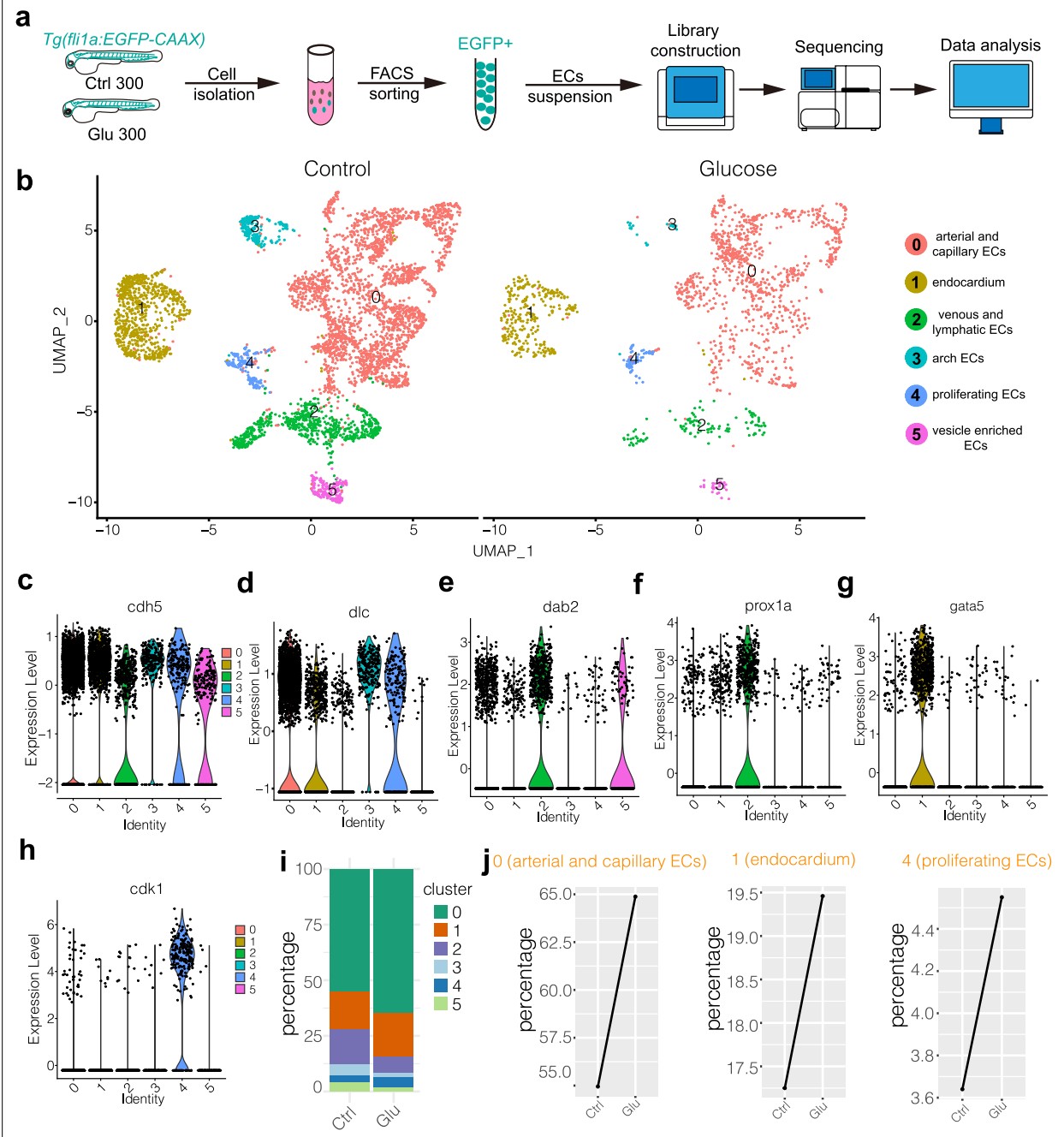

**Figure 4.** Single-cell transcriptome sequencing analysis of endothelial cells (ECs) in control and high glucose-treated embryos. (**a**) Schematic diagram of the single-cell sequencing process. 300 embryos in the control group and 300 in the high glucose group were used, and ECs were sorted by GFP fluorescent using fluorescence-activated cell sorting (FACS) technology. (**b**) The measured cells were divided into six individual clusters based on gene expression profiles using UMAP. (**c–h**) The violin plots of some EC marker genes. (**i**) The proportion of ECs in each cluster of control and high glucose groups. (**j**) Changes of ECs percentage in arterial and capillary ECs, endocardium, and proliferating ECs of control and high glucose group.

The online version of this article includes the following figure supplement(s) for figure 4:

**Figure supplement 1.** Overview of the number of genes, total Unique Molecular Identifiers (UMIs), and percentage of mitochondrial UMIs for the single-cell RNA sequencing.

**Figure supplement 2.** UMAP representation of endothelial cell (EC) subpopulations.

**Figure supplement 3.** Transcriptome sequencing analysis of control, high D-glucose-, and high L-glucose-treated embryos.

**Figure supplement 4.** The heatmap of metabolism-related genes.

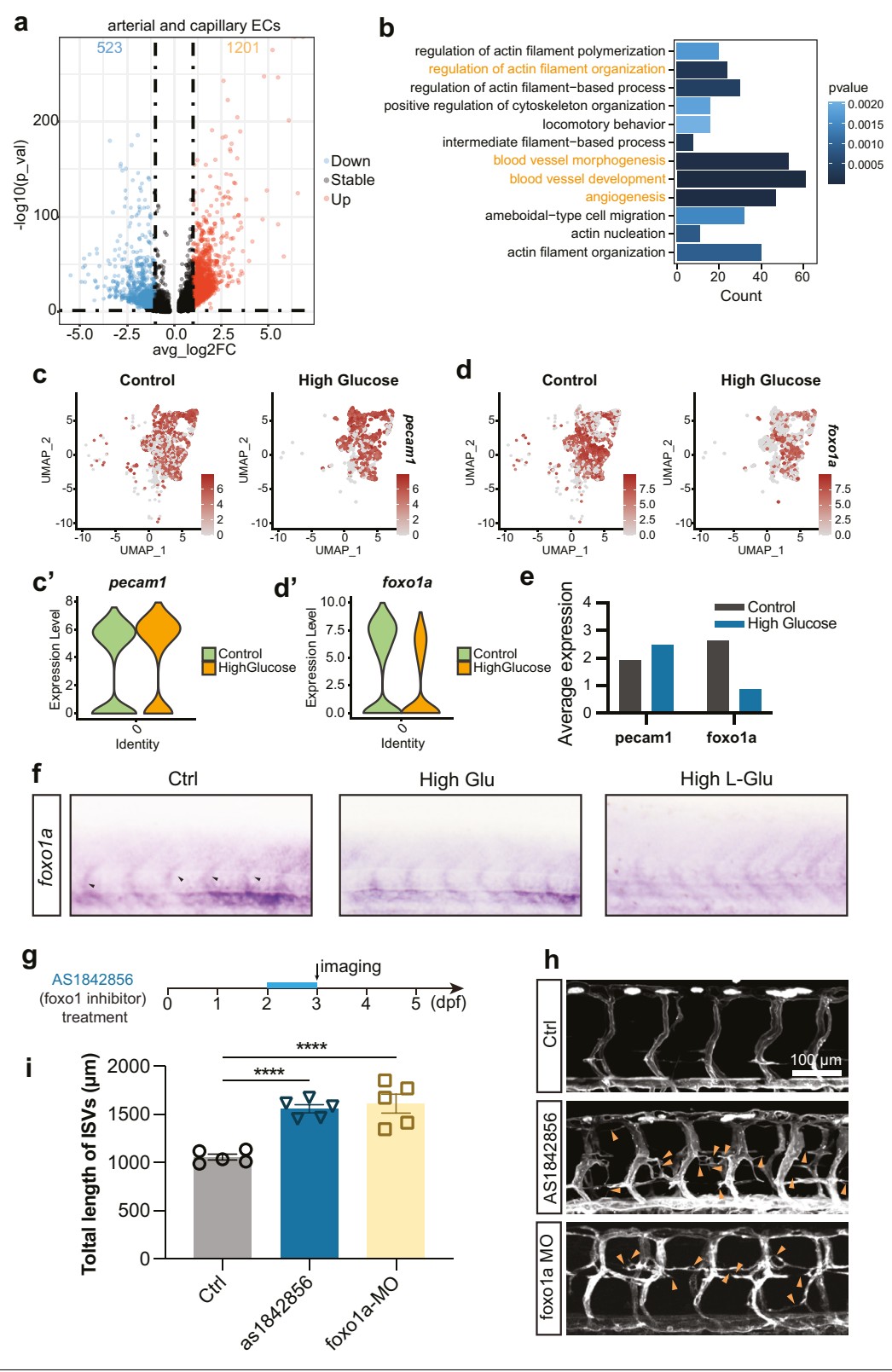

**Figure 5.** Foxo1a was involved in the excessive angiogenesis induced by high glucose treatment. (**a**) The volcano plot of differential expression genes in arterial and capillary endothelial cells (ECs). The avg_log2FC greater than 1 was considered significant, including 523 downregulated genes (blue dots) and 1201 upregulated genes (red dots). (**b**) Gene ontology (GO) analysis of 523 downregulated genes in arterial and capillary ECs. (**c**) The feature plot of

*Figure 5 continued on next page*

*Figure 5 continued*

ECs marker gene pecam1 of control and high glucose group in arterial and capillary ECs. (**c'**) The violin plot of ECs marker gene pecam1 of control and high glucose group in arterial and capillary ECs. (**d**) The feature plot of gene foxo1a of control and high glucose group in arterial and capillary ECs. (**d'**) The violin plot of gene foxo1a of control and high glucose group in arterial and capillary ECs. (**e**) Average expression of gene pecam1 and foxo1a in control and high glucose group. (**f**) Whole-mount in situ hybridization analysis of foxo1a in control, high glucose-, and high L-glucose-treated embryos. (**g**) A diagram showing the foxo1 inhibitor treatment time window. (**h**) Confocal imaging analysis of control embryos, AS1842856-treated embryos, and foxo1a MO-injected embryos. Arrowheads indicate ectopic angiogenic sprouts. (**i**) Statistical analysis of the total length of intersegmental vessels (ISVs) in control embryos, AS1842856-treated embryos, and foxo1a MO-injected embryos (n=5). t-test, ****p<0.0001.

gene of Foxo1a. Therefore, we investigated the impact of Foxo1 inhibition on *marcksl1a* expression in zebrafish embryos. As expected, qPCR analysis revealed that inhibition of Foxo1a by AS1842856 resulted in the upregulation of marcksl1a expression. In contrast, Foxo1a overexpression resulted in the downregulation of *marcksl1a* (*Figure 8a–c*), which suggested that Foxo1a might negatively regulate *marcksl1a* transcription in zebrafish. To further confirm it, we performed the chromatin immunoprecipitation (ChIP) experiment to validate the potential binding interaction between Foxo1 and *marcksl1a*. Since the amino acid sequence and DNA binding motifs of Foxo1 are highly conserved between zebrafish and mice (*Figure 7—figure supplement 1*), we analyzed the 3 kb promoter region of *marcksl1a* to search the binding site (BS) sequence of mouse FOXO1 presented in the JASPAR database. Two candidate BS were found at −265 to −275 (BS1) and −153 to −163 (BS2) nucleotides upstream of the TSS of *marcksl1a* (*Figure 8d*) and then used for the ChIP-PCR assay detection. The results showed that Foxo1a was enriched in both the predicted binding sites of *marcksl1a* (*Figure 8e*) in zebrafish. Luciferase reporter assay also indicated that Foxo1a could negatively regulate marcksl1a transcription in zebrafish (*Figure 8f and g*).

Additionally, we microinjected *marcksl1a* MO into the one-cell stage *Tg(fli1a:EGFP-CAAX)* [ntu666] embryos, which were then treated with high levels of D-glucose and L-glucose. The findings revealed that the knockdown of Marcksl1a could effectively mitigate the excessive angiogenesis caused by high D-glucose or high L-glucose treatment, resembling the rescue effect observed with VEGFR inhibitor Lenvatinib (*Figure 8h–n*, *Figure 7—figure supplement 2*). These results suggested that monosaccharides induced excessive angiogenesis through the Foxo1a-*marcksl1a* pathway in zebrafish embryos.

## Discussion

In this study, we successfully established a new zebrafish model with significant excessive angiogenesis, resembling the hyperangiogenic characteristics observed in PDR more closely than previously established models (*Jung et al., 2016*; *Jörgens et al., 2015*). Jung et al. have described a short-term zebrafish model for diabetic retinopathy (DR) induced by high glucose, which exhibited blood vessel defects (*Jung et al., 2016*). However, these defects were limited to the disruption of tight junctions and dilation of hyaloid-retinal vessels (*Jung et al., 2016*), without the excessive angiogenesis and vascular blockage observed in PDR and our established model. Additionally, although Jörgens et al. have observed the hyperbranching of small vessel structures originating from the upper part of ISVs, growing horizontally toward and partially connecting to the neighboring ISVs field (*Jörgens et al., 2015*), the angiogenic sprouts did not form a more complex structure that was observed in our research.

The excessive development of immature blood vessels represents a significant pathological condition in the progression of DR and nephropathy (*Wilkinson-Berka, 2004*; *Osterby and Nyberg, 1987*). Hyperglycemia has been considered one of the most causal factors causing vascular damage, including excessive angiogenesis. However, the exact mechanism through which hyperglycemia impairs the blood vessels is not well determined. We analyzed single-cell transcriptomic sequencing data of the ECs isolated from D-glucose-treated embryos to gain more insights into it. The findings revealed an increased ratio of tip cells and proliferating ECs, accompanied by the altered expression of various angiogenic genes in the ECs of D-glucose-treated embryos.

Foxo1 has been validated to be essential for sustaining the quiescence of ECs, with involvement in metabolism regulation (*Wilhelm et al., 2016*; *Andrade et al., 2021*). Moreover, it also plays an important role in diabetic microvascular complications, including DR (*Parmar et al., 2023*; *Behl et al.,*

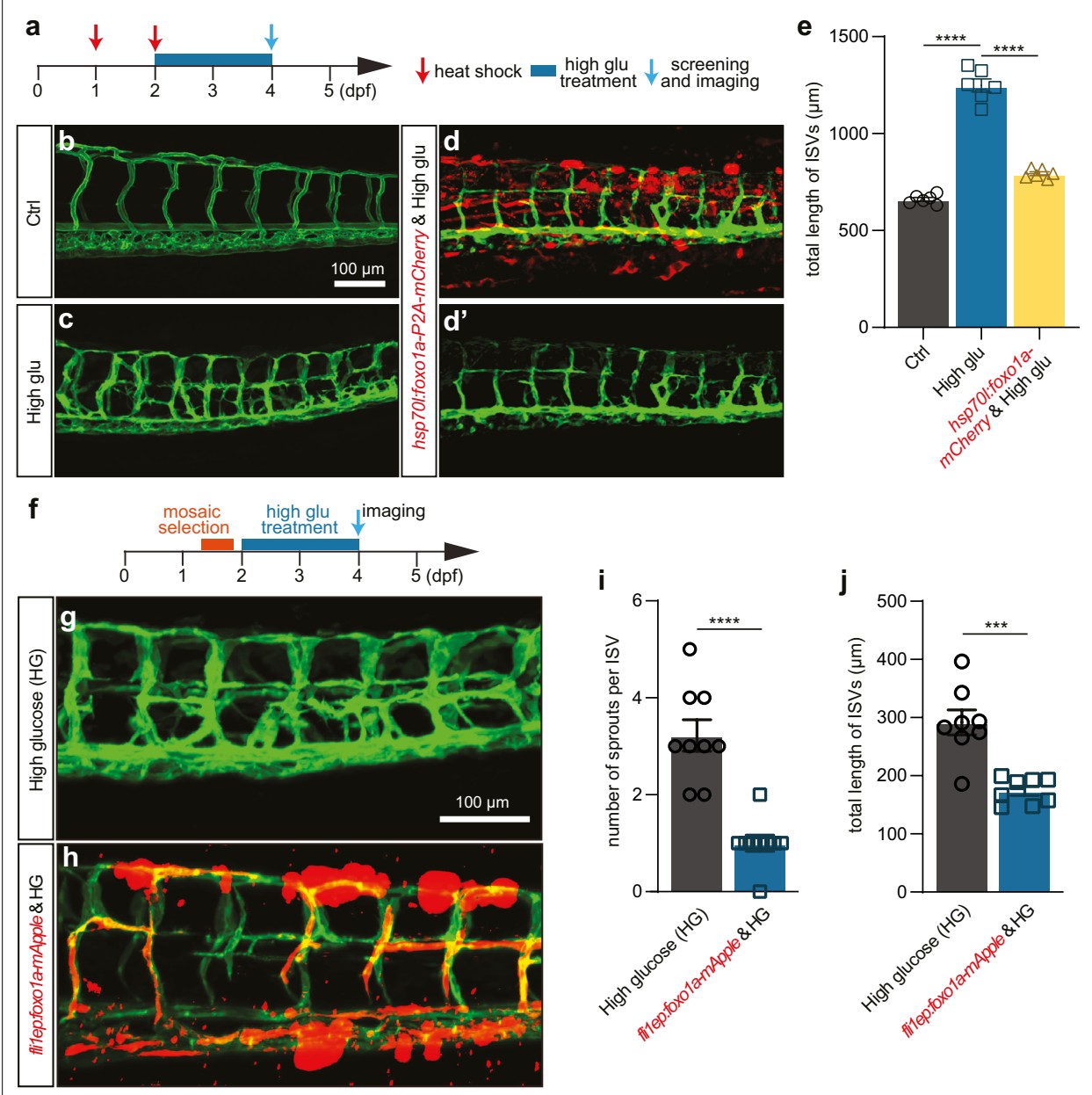

**Figure 6.** Foxo1a gain of function can partially rescue the excessive angiogenesis induced by high glucose treatment. (**a**) A diagram showing the high glucose treatment time window and heat shock-treated time point. (**b**) Confocal imaging analysis of blood vessels in control *Tg(fli1aEP:EGFP-CAAX)^{ntu666}* embryos. (**c**) Confocal imaging analysis of blood vessels in high glucose-treated *Tg(fli1aEP:EGFP-CAAX)^{ntu666}* embryos. (**d–d'**) Confocal imaging analysis of blood vessels in *hsp70l:foxo1a-P2A-mCherry* and high glucose-treated *Tg(fli1aEP:EGFP-CAAX)^{ntu666}* embryos. (**e**) Statistical analysis of the total length of intersegmental vessels (ISVs) in control, high glucose-treated, *hsp70l:foxo1a-P2A-mCherry, and high glucose*-treated embryos (n=6), respectively. One-way ANOVA, ****$p < 0.0001$. (**f**) A diagram showing the high glucose treatment time window and imaging time point. (**g**) Confocal imaging analysis of blood vessels in high glucose-treated *Tg(fli1aEP:EGFP-CAAX)^{ntu666}* embryos. (**h**) Confocal imaging analysis of blood vessels in *fli1EP:foxo1a-mApple* and high glucose-treated *Tg(fli1aEP:EGFP-CAAX)^{ntu666}* embryos. (**i**) Statistical analysis of the number of sprouts per ISV in high glucose-treated embryos, and *fli1EP:foxo1a-mApple* and high glucose-treated embryos (n=9). t-test, ****$p < 0.0001$. (**j**) Statistical analysis of the total length of ISVs in high glucose-treated embryos, and *fli1EP:foxo1a-mApple* and high glucose-treated embryos (n=8). t-test, ***$p < 0.001$.

*2022*). Here, by combining the single-cell transcriptomic sequencing data analysis and experimental validation, we identified the transcription factor Foxo1a, which was significantly downregulated in the embryos treated with high glucose, responsible for the excessive angiogenesis. Additionally, our result further revealed that Foxo1a exerts its regulatory function during this process by downregulating its target gene *marcksl1a*, regardless of whether the embryos were treated with D-glucose or

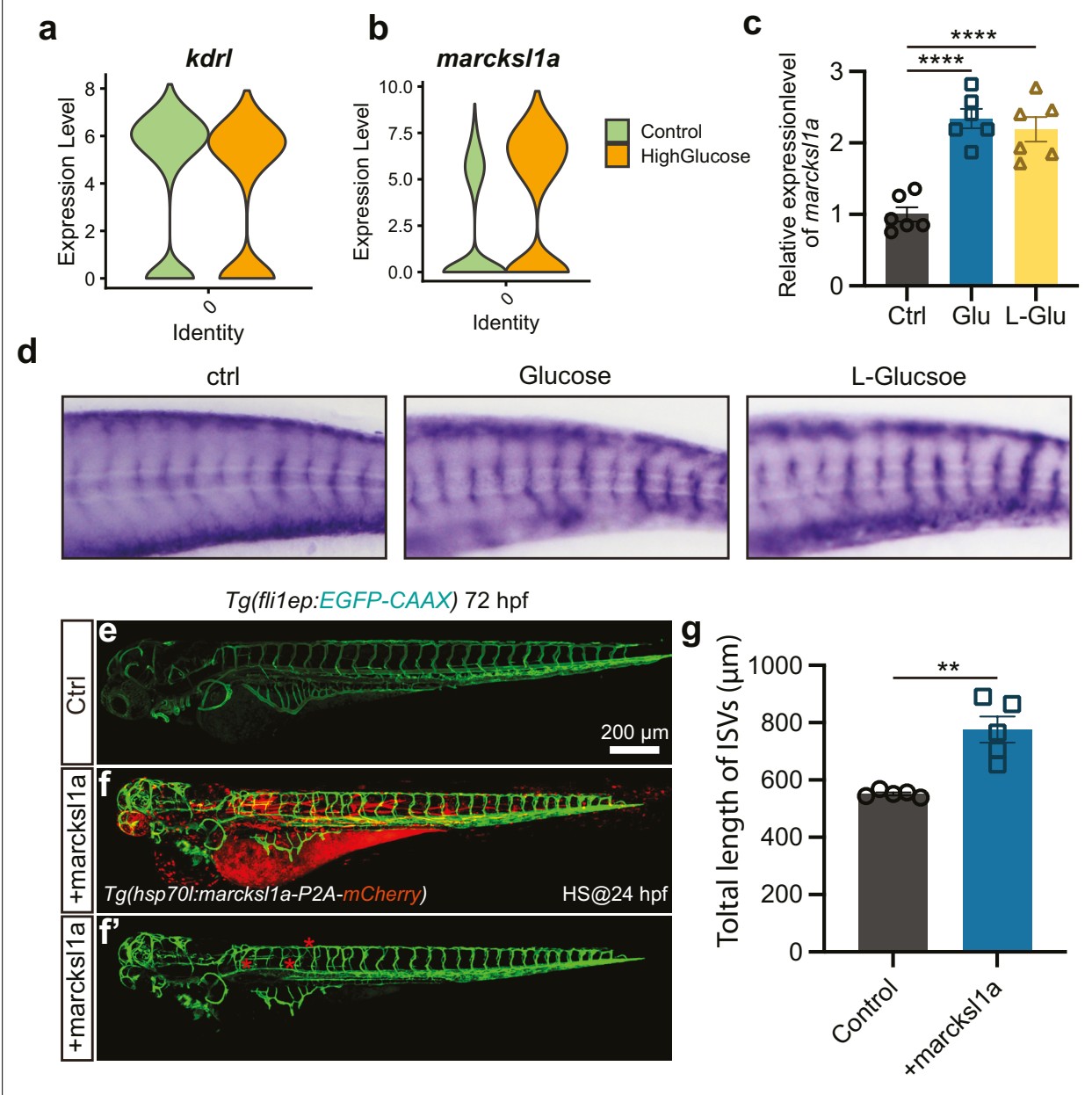

**Figure 7.** Marcksl1a overexpression induced excessive angiogenesis in zebrafish embryos. (**a**) The violin plot of endothelial cells (ECs) marker gene *kdrl* of control and high glucose group in arterial and capillary ECs. (**b**) The violin plot of gene *marcksl1a* of control and high glucose group in arterial and capillary ECs. (**c**) Real-time PCR analysis of *marcksl1a* expression in control, high glucose-, and high L-glucose-treated embryos (n=6). t-test, ****p<0.0001. (**d**) Whole-mount in situ hybridization analysis of *marcksl1a* in control, high glucose, and high L-glucose-treated embryos. (**e–f'**) Confocal imaging analysis of blood vessels in control and *hsp70l:marcksl1a-P2A-mCherry*-injected *Tg(fli1aEP:EGFP-CAAX)^{ntu666}* embryos. (**g**) Statistical analysis of the total length of intersegmental vessels (ISVs) in control and *hsp70l:marcksl1a-P2A-mCherry*-injected embryos (n=5). t-test, **p<0.01.

The online version of this article includes the following figure supplement(s) for figure 7:

**Figure supplement 1.** Multiple amino acid sequence alignment of mouse FOXO1 and zebrafish Foxo1a.

**Figure supplement 2.** Confocal imaging analysis of blood vessels in the embryos with Lenvatinib treatment.

L-glucose. Taken together, our results suggested that both caloric and noncaloric monosaccharides treatment could lead to excessive angiogenesis by promoting the differentiation of quiescent ECs into tip cells through the *foxo1a-marcksl1a* pathway.

Previous studies have linked the consumption of ASB to the occurrence and development of cardiovascular disease (*Fung et al., 2009*; *de Koning et al., 2012*; *Fagherazzi et al., 2013*; *Gardener*

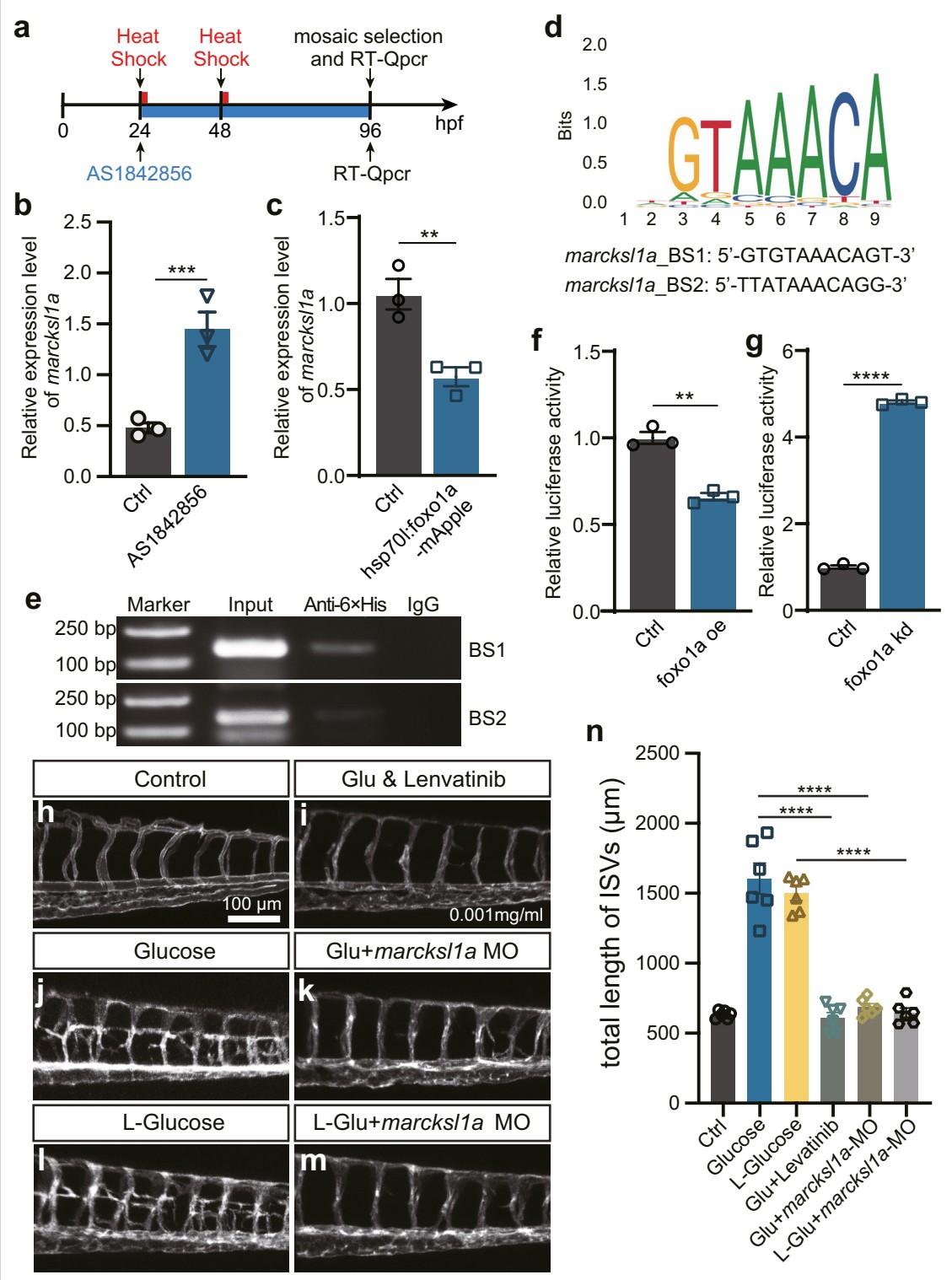

**Figure 8.** Noncaloric monosaccharides induced excessive angiogenesis through foxo1a-marcksl1a signal in zebrafish embryos. (**a**) A diagram showing the Foxo1 inhibitor and heat shock treatment time window. (**b**) Real-time PCR analysis of *marcksl1a* expression in control and AS1842856-treated embryos (n=3). t-test, ***p<0.001. (**c**) Real-time PCR analysis of *marcksl1a* expression in control and *foxo1a* overexpressed embryos (n=3). t-test, **p<0.01. (**d**) A sequence logo of Foxo1-binding sequence presented in the JASPAR database (https://jaspar.genereg.net/) and two candidate binding sites at the upstream of transcription start site (TSS) of *marcksl1a* in zebrafish. (**e**) Results of the chromatin immunoprecipitation (ChIP)-PCR assay indicated that BS1 and BS2 are Foxo1a-binding sites of *marcksl1a* in zebrafish. Input sonicated genomic DNA samples without immunoprecipitation as a positive control. IgG, sonicated, and IgG-immunoprecipitated genomic DNA samples as a negative control. (**f, g**) Luciferase reporter activity in *foxo1a*

*Figure 8 continued on next page*

*Figure 8 continued*

overexpressed or knocked down embryos (n=3), respectively. t-test, **p<0.01, ****p<0.0001. (**h–m**) Confocal imaging analysis of blood vessels in control, high glucose, high glucose and Lenvatinib, high glucose+*marcksl1a* MO, high L-glucose, and high L-glucose+*marcksl1a* MO groups. (**n**) Statistical analysis of the total length of intersegmental vessels (ISVs) in the groups in h–m (n=6), respectively. One-way ANOVA, ****p<0.0001.

*et al., 2012*; *Drouin-Chartier et al., 2019*; *de Koning et al., 2011*; *Hirahatake et al., 2019*; *Mossavar-Rahmani et al., 2019*; *Vyas et al., 2015*). However, the potential mechanisms underlying the association have not been well documented. In recent years, positive associations between ASB and cardiovascular disease have been proposed, possibly due to several plausible factors, including the potential impact of ASBs on central nervous system circuits, gut hormone secretion, and gut microbiota (*Blundell and Hill, 1986*; *Pepino, 2015*; *Nettleton et al., 2016*). Additionally, it has been hypothesized that the ASBs might stimulate appetite and increase calorie intake (*Blundell and Hill, 1986*; *Pepino, 2015*).

For a long time, there has been considerable debate and conflicting opinions regarding how specific sugars affect the development of type 2 diabetes rather than excess calories per se (*Qi and Tester, 2020*; *Laville and Nazare, 2009*). In this study, we have provided new evidence indicating that the administration of noncaloric monosaccharides leads to significant excessive angiogenesis, suggesting that the excessive angiogenesis may not be only attributed to the caloric properties. Since excessive angiogenesis is the major pathological feature of DR and nephropathy, our findings are in support of a possible biological mechanism underlying the positive associations between noncaloric monosaccharides and microvascular complications associated with type 2 diabetes, suggesting that the noncaloric monosaccharides might not be suitable for ASB consumption.

Surprisingly, no notable abnormalities were observed in the vessels of embryos treated with disaccharides, including lactose, maltose, and sucrose, which is consistent with the previous study stating that intakes of sucrose, lactose, and maltose were not significantly associated with the risk of type 2 diabetes (*Montonen et al., 2007*). This finding implied that the effects induced by monosaccharides cannot be attributed to the osmotic pressure of the surrounding medium. Furthermore, despite the potential conversion of these disaccharides into monosaccharides, the restricted reaction rate may maintain them within a safe concentration range that is not harmful to the vessels in a short period.

In conclusion, to investigate the effects of monosaccharides on vascular development, we established a zebrafish model by treating the embryos with high concentrations of monosaccharides. Based on this model, we observed significant excessive angiogenesis induced by glucose and noncaloric monosaccharides, initiated by activating the quiescent ECs into proliferating tip cells. The effects of monosaccharides on inducing excessive angiogenesis were then proved to be mediated by the *foxo1a-marcksl1a* pathway. The results have provided novel insights into the roles of noncaloric monosaccharides in human health and the underlying mechanisms.

## Materials and methods

### Zebrafish

Care and breeding of zebrafish were carried out as previously described (*Wang et al., 2016*). Animal experiments were conducted according to local institutional laws and Chinese law for the Protection of Animals. The following transgenic strains were used: *Tg(fli1aEP:EGFP-CAAX)^ntu666^* and *Tg(kdrl:ras-mCHerry)* (*Krueger et al., 2011*). Embryos were obtained through natural mating and maintained at 28.5°C. The stages of zebrafish embryos are defined as previously described (*Wang et al., 2016*). Embryos were treated with 0.2 mM 1-phenyl-2-thiourea (PTU, Sigma, P7629) to block pigmentation for further imaging analysis.

### Monosaccharides and drug treatment

The D-glucose (Sigma, G7021-100g), L-glucose (Sigma, G5500-1g; J&K, 981195-1g), D-fructose (Sigma, F0127-100g), L-rhamnose monohydrate (Aladdin, R108982), D-sorbitol (Sigma, S1876-100g), D-mannitol (Sigma, M4125-100g), D-(-)-ribose (Sigma, V900389-25g), L-(+)-arabinos (Sigma, V900920-25g), mannose (Sigma, M2069-25g), and sucrose (Sigma, V900116-500G) were dissolved in E3 solution. Zebrafish embryos at 24 hpf to 48 hpf were placed in 24-well plates (10 embryos per well) and immersed in the solution at the presetting concentrations and time windows. Then, put it

in a 28°C incubator for cultivation. Five days before embryonic development, a stereo fluorescence microscope and a laser confocal microscope were used to observe the changes in blood vessel phenotype. For the drug treatment, the embryos were co-incubated in glucose with Lenvatinib (Selleck, S1164-5MG) at the presetting concentrations and time window. Foxo1 inhibitor AS1842856 (MCE, HY-100596) was dissolved in DMSO and stored at –80°C and diluted with E3 solution to 1 μM when used. The same concentration of DMSO was used as a negative control.

## Glucose concentration measurement

Glucose concentration in the embryo was measured as described previously (*Da'as et al., 2023*). Embryos that developed to 75% epiboly were selected and transferred to 24-well plates (10 embryos per well) and immersed in the solution at the presetting concentrations and time windows. For glucose concentration measurement, embryos (n=20) were transferred to a new 1.5 mL tube, rinsed three times with 1× PBS, and immersed in ice for the following experiments. Discard the PBS as much as possible, embryos homogenized using a hand homogenizer, and centrifuged at 14,000×*g* for 10 min. 1.5 μL of the supernatant was used to measure the total free-glucose level using a glucometer (Baye, 7600P).

## Whole-mount ISH

Whole-mount ISH and the preparation of antisense RNA probes were performed as described in the previous protocol (*Huang et al., 2013*). Briefly, the *marcksl1a* and *foxo1a* cDNA fragments were cloned with the specific primers listed below using the wild-type embryo (AB) cDNA library. Probes were synthesized using the in vitro DIG-RNA labeling transcription Kit (Roche, 11175025910) with linearized pGEM-T easy vector containing *marcksl1a* or *foxo1a* gene fragment as the templates. Synthesized probes were purified with LiCl (Invitrogen, AM9480) and diluted to 1 ng/μL for hybridization. Zebrafish embryos were collected and fixed with 4% paraformaldehyde overnight at 4°C, then dehydrated with methanol gradients and stored at –20°C in 100% methanol. The hybridization result was detected with anti-DIG-AP antibody (1:2000, Roche, 11093274910) and NBT/BCIP (1:500, Roche, 11681451001). After hybridization, images of the embryos were captured with an Olympus stereomicroscope MVX10. The primers are listed below:

> *marcksl1a*-probe-forward:5'- AGG ATG GGT GCT CAG TTG AC-3'
> *marcksl1a*-probe-reverse:5'- GCT GGC GTC TCA TTG GTT TC-3'
> *foxo1a*-probe-forward:5'-GCA ACA CAG GAT TTC CCC AC-3'
> *foxo1a*-probe-reverse:5'-CAC AGG TGG CAC TGG AAG G-3'

## Single-cell gene expression profile analysis

Cell Ranger 3.0.2 (https://github.com/10XGenomics/cellranger; *10xGenomics Company, 2019*) was used to convert the raw sequencing data to a single-cell level gene count matrix. The clustering of single cells and the marker genes in each cluster were analyzed by Seurat 3.0 (https://satijalab.org/seurat/install.html; *Stuart et al., 2019*). Several criteria were applied to select the single cells, including only keeping the genes that are expressed (UMI larger than 0) at least in three single cells, selecting single cells with the number of expressed genes at the range between 500 and 3000, and requiring the percentage of sequencing reads on mitochondrial genome being less than 5 percentage. Furthermore, sctransform method (*Hafemeister and Satija, 2019*) was applied to remove technical variation, and ClusterProfiler (*Yu et al., 2012*; *Yu et al., 2019*) was used to do the GO enrichment analysis based on the marker genes of each cell cluster. Detailed information about the data processing can be found in this project's source code (https://github.com/gangcai/ZebEndoimmune; *Xie, 2024*).

## Gene expression analysis by real-time qPCR

Total RNA was extracted from zebrafish embryos using TRIzol (Invitrogen, 15596026) and stored at –80°C. The cDNA was then synthesized using the HiScript II Q RT SuperMix for qPCR Kit (Vazyme, R223-01) according to the manufacturer's instructions. qPCR was performed in triplicates using the Taq Pro Universal SYBR qPCR Master Mix (Vazyme, Q712-02) on a real-time PCR detection system (StepOne Real-Time PCR Systems). The primers used for real-time PCR analysis are as follows:

> *ef1α*-Qpcr-F:5'- CTT CAA CGC TCA GGT CAT CA -3'

*ef1α*-Qpcr-R:5'- CGG TCG ATC TTC TCC TTG AG -3
*marcksl1a*-Qpcr-F:5'- CCG TGG CTG ATA AAG CCA AT -3'
*marcksl1a*-Qpcr-R:5'- CTC CCT CCT CCG TTT TTG GG -3'

## Transgenic and heat shock

The *Tg(fli1aEP:EGFP-CAAX)*[ntu666] line was established using a construct *fli1aEP:EGFP-CAAX*, which was generated using MultiSite Gateway technology, the tol2 kit as previously described (**Kwan et al., 2007**). The 5' Entry p5Efli1ep (#31160) purchased from Addgene was originally from Nathan Lawson Lab (**Villefranc et al., 2007**). Three entry clones and the pDestTol2pA2 destination vector were used to generate the expression construct by LR recombination reaction as described in the Multisite Gateway Manual book. The expression constructs were synthesized by GENEWIZ company. The zebrafish embryos were immersed in a 37°C water bath for 1 hr for heat shock. Around 75 pg of expression plasmid DNA and 25 pg tol2 transposase mRNA were premixed and microinjected into single-cell fertilized eggs.

## ChIP-PCR

Embryos injected with *hsp70l:foxo1a-6×His-P2A-mCherry* were collected at 72 hpf after heat shock treatment. According to the manufacturer's instructions, the ChIP-PCR assay was performed using the ChIP Assay Kit (Millipore, 3753379). The genomic DNA crossed with Foxo1a protein was immunoprecipitated by using 5 μg Anti-6×His tag antibody (abcam, ab213204). Antibody against IgG was used as a negative control. The semiquantitative PCR was performed with KODfx (TOYOBO, KFX-101) at the following conditions: 94°C for 5 min; 35 cycles of 98°C for 10 s, 55°C for 30 s, 68°C for 10 s; 68°C for 10 min. The PCR primers used for the predicted BS are as follows:

*marcksl1a*-BS1-forward:5'- CCC TTT TTC AAA AGT GAG TTT GAG -3'
*marcksl1a* -BS1-reverse:5'- GGA GCT TCA TCT GCC CCA TT -3'
*marcksl1a* -BS2-forward:5'- CGG TTT CCA GCT TTC TTC AGA A 3'
*marcksl1a* -BS2-reverse:5'- TCT CAA ACT CAC TTT TGA AAA AGG G -3'

## Luciferase reporter assay

Plasmids of PGL4.10[luc2] and PGL4.74[hRluc/TK] (Promega) were used for luciferase reporter assay. Zebrafish *marcksl1a* promoter fragment with predictive Foxo1a-binding cite was cloned and inserted into pGL4.10 basic vector by Kpn I and Hind III. The assays for detecting the promoter activity in response to Foxo1a were performed according to the previous study. Briefly, 50 pg of PGL4.10 vectors, 1 pg of PGL4.74 vectors, and 50 pg of *foxo1a* mRNA or 20 pg of *foxo1a* MO were co-injected into wild-type embryos at the one-cell stage. And then, embryos were harvested at 24 hpf to measure their luciferase activity according to the manufacturer's protocols (Promega).

## Imaging analysis

For confocal imaging of blood vessels in fluorescence protein labeled transgenic zebrafish embryos, they were anesthetized with egg water/0.16 mg/mL MS222 (Sigma, A5040)/1% PTU and embedded in 0.5–0.8% low melting agarose. Confocal imaging was performed with a Nikon A1R HD25 Confocal Microscope. Analysis was performed using Nikon-NIS-Elements software. The bright-field images were acquired with an Olympus DP71 camera on an Olympus stereomicroscope MVX10.

## Statistical analysis

Statistical analysis was performed with a Student's t-test. All data is presented as mean ± SEM; $p < 0.05$ was considered statistically significant.

## Acknowledgements

This study was supported by grants from the National Natural Science Foundation of China (81870359, 2018YFA0801004).

## Additional information

### Funding

| Funder | Grant reference number | Author |
|---|---|---|
| National Natural Science Foundation of China | 2018YFA0801004 | Dong Liu |
| National Natural Science Foundation of China | 81870359 | Dong Liu |

The funders had no role in study design, data collection and interpretation, or the decision to submit the work for publication.

### Author contributions

Xiaoning Wang, Data curation, Formal analysis, Visualization, Writing – original draft; Jinxiang Zhao, Data curation, Formal analysis, Visualization; Jiehuan Xu, Data curation, Formal analysis; Bowen Li, Resources, Software; Xia Liu, Project administration, Writing – review and editing; Gangcai Xie, Resources, Software, Project administration; Xuchu Duan, Visualization, Writing – original draft, Project administration, Writing – review and editing; Dong Liu, Conceptualization, Funding acquisition, Writing – original draft, Project administration, Writing – review and editing

### Author ORCIDs

Xiaoning Wang ⓘ https://orcid.org/0000-0003-0222-0755
Xia Liu ⓘ http://orcid.org/0000-0001-5473-5596
Gangcai Xie ⓘ http://orcid.org/0000-0002-8286-2987
Xuchu Duan ⓘ https://orcid.org/0000-0002-4763-126X
Dong Liu ⓘ https://orcid.org/0000-0002-2764-6544

### Ethics

All zebrafish experimentation was carried out following the NIH Guidelines for the care and use of laboratory animals (http://oacu.od.nih.gov/regs/index.htm) and ethically approved by the Administration Committee of Experimental Animals, Jiangsu Province, China (Approval ID: 20180905-002).

Reviewer #1 (Public review): https://doi.org/10.7554/eLife.95427.3.sa1
Reviewer #2 (Public review): https://doi.org/10.7554/eLife.95427.3.sa2
Reviewer #3 (Public review): https://doi.org/10.7554/eLife.95427.3.sa3
Author response https://doi.org/10.7554/eLife.95427.3.sa4

## Additional files

### Supplementary files

• Supplementary file 1. Cell proportions and numbers of each cluster.

• Supplementary file 2. Cell proportions and numbers of each cluster.

• MDAR checklist

### Data availability

Sequencing data have been deposited in GEO under accession codes GSE276251 and GSE276252.

The following datasets were generated:

| Author(s) | Year | Dataset title | Dataset URL | Database and Identifier |
|---|---|---|---|---|
| Wang X, Zhao J, Xu J, Li B, Liu X, Xie G, Duan X, Liu D | 2024 | Single-cell transcriptome sequencing analysis of endothelial cells in control and high glucose treated zebrafish embryos | https://www.ncbi.nlm.nih.gov/geo/query/acc.cgi?acc=GSE276251 | NCBI Gene Expression Omnibus, GSE276251 |

*Continued on next page*

*Continued*

| Author(s) | Year | Dataset title | Dataset URL | Database and Identifier |
|---|---|---|---|---|
| Wang X, Zhao J, Xu J, Li B, Liu X, Xie G, Duan X, Liu D | 2024 | Effect of D-glucose and L-glucose on gene expression of zebrafish embryos at 72 hpf (hours post fertilization) and 96 hpf | https://www.ncbi.nlm.nih.gov/geo/query/acc.cgi?acc=GSE276252 | NCBI Gene Expression Omnibus, GSE276252 |

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
