## [Editor Report · eLife assessment]

This **valuable** study investigates the effect of noncaloric monosaccharides, sugar substitutes that are commonly used by diabetic patients, on angiogenesis in the zebrafish embryo. The authors show that noncaloric monosaccharides and glucose similarly induce excessive blood vessel formation due to the increased formation of tip cells by endothelial cells through the foxo1a-marcksl1a pathway. This **solid** study is of interest for the medical community in charge of the prevention and of the treatment of diabetes and other metabolic diseases.

---

## [Referee Report · Reviewer #1 (Public review)]

Dong Liu et al. successfully established a short-term zebrafish model by treating the embryos with high concentrations of monosaccharides, resembling the hyperangiogenic characteristics observed in proliferative diabetic retinopathy. The authors found that excessive angiogenesis induced by glucose and noncaloric monosaccharides can be achieved by activating the quiescent endothelial cells into proliferating tip cells. Importantly, the authors further confirmed the effects of monosaccharides on inducing excessive angiogenesis were mediated by the foxo1a-marcksl1a pathway. These results demonstrate the potentially detrimental effects of the noncaloric monosaccharides on blood vessel function and provided novel insights into the underlying mechanisms.

---

## [Referee Report · Reviewer #2 (Public review)]

In the manuscript Liu et al. observed that glucose and noncaloric monosaccharides can prompt an excessive formation of blood vessels, particularly intersegmental vessels (ISVs). They propose that these branched vessels arise from the ectopic activation of quiescent endothelial cells (ECs) into tip cells. Moreover, through single-cell transcriptome sequencing analysis of embryonic endothelial cells exposed to glucose, they noted an increased proportion of arterial and capillary endothelial cells, proliferative endothelial cells, along with a series upregulated genes in categories of blood vessel morphogenesis, development, and pro-angiogenesis. The authors provide evidence suggesting that caloric and noncaloric monosaccharides (NMS) induce excessive angiogenesis via the Foxo1a-Marcksl1a pathway.

---

## [Referee Report · Reviewer #3 (Public review)]

The authors have investigated the effect of noncaloric monosaccharides on angiogenesis in the zebra fish embryo. These compounds are used as substitutes of sugars to sweeten beverages and they are commonly used by diabetic patients. The authors show that noncaloric monosaccharides and glucose similarly induce excessive blood vessels formation due to increased formation of tip cells by endothelial cells. The authors show that this excessive angiogenesis involved the foxo1a-marcksl1a pathway.

A limitation of the study is that the mechanism of angiogenesis in the retinal circulation and in peripheral vasculature is certainly different.

This result suggests that these noncaloric monosaccharides share common side effects with glucose. Consequently, more caution should be taken as regard to the use of these artificial sweeteners. This work is of interest for a better management of diabetes.

---

## [Author Response]

The following is the authors’ response to the original reviews.

**Reviewer #1:**
(1) The authors claimed that they examined the arterial and venous identity of the hyperbranched vessels via live imaging analysis of the high glucose-treated Tg(flt1:YFP::kdrl:ras-mCherry) line, and revealed that the hyperbranched ectopic vessels comprised arteries and veins. That's good, of course. However, there are no relevant results in Figure 2. Please revise it.

Thank you very much for the suggestion. We’ve added this part of the results in Figure 2i and j.

(2) In Figures 3f and 3g, some of the ECs protruded long and intricate sprouts, and nearly all the ECs within an ISV underwent the outgrowth of filopodia in some extreme cases (Figure 3g), suggesting that the high glucose treatment induced the endothelial differentiation into tip cell-like cells. The findings are surprising and interesting. In order to further confirm the author's conclusion, in situ hybridization experiments are more appropriate to show the expression changes of tip cell-like cell marker genes in the high glucose-treated embryos.

Thank you very much for your constructive suggestions. We have performed the analysis of single-cell RNA-seq data, and the results showed that the tip cell marker genes such as *esm1*, *apln*, and *cxcr4a* were significantly up-regulated in arterial and capillary ECs after high glucose treatment. The results were integrated into Figure 3 of the revised manuscript.

(3) Embryos treated with AS1842856 or injected with foxo1a-MO exhibited excessive angiogenesis (Figure 5g-i), suggesting the transcription activity of foxo1 is required to maintain the quiescent state of endothelial cells. Did the downregulation of foxo1a lead to the differentiation of endothelial cells into tip-cell-like cells?

Thank you very much for the question. We examined our results carefully and marked these tip cell-like cells with arrow heads in Figure 5h of the revised manuscript.

(4) Foxo1a was significantly downregulated in arterial and capillary ECs after high glucose treatment (Figure 5c-e). More importantly, whether overexpression of foxo1a in the high glucose-treated embryos could eliminate the hyperangiogenic characteristics?

Thank you for the great questions. We performed rescue experiments, and the results suggested that the overexpression of *foxo1a* partially mitigated the excessive angiogenesis induced by high glucose treatment. These results were integrated into Figure 6 of the revised manuscript.

(5) The authors' results found that foxo1a was enriched in both the predicted binding sites of marcksl1a by ChIP-PCR experiments (Figure 7d). This result is reliable. However, whether these two sites are important for marcksl1a gene transcription needs to be confirmed by relevant experiments, such as luciferase reporter assays.

We’ve performed the luciferase reporter assays and added these data to Figure 8f and g.

**Reviewer #2:**
Suggested major experiments:(1) A previous study (Jorgens et al., Diabetes 64, 2015) reported that high tissue glucose levels increased reactive dicarbonyl methylglyoxal (MG) concentrations in zebrafish embryos and triggered the formation of hyperbranched ISVs. Additionally, they illustrated that MG induced the vascular hyperbranching phenotype via enhancing phosphorylated VEGFR and pAKT signaling cascade. The authors must examine whether both pVEGFR and pAKT are increased in noncaloric monosaccharide (NMS)-treated embryos. The authors need also to test the crosstalks between VEGFR/AKT signaling and foxo1a-Marcksl1a pathway in glucose or NMS-treated embryos.

Thank you very much for your suggestion. We treated the embryos with AS1842856 (foxo1 inhibitor) and Lenvatinib (VEGFR inhibitor), and the results showed that Lenvatinib treatment attenuated the excessive angiogenesis induced by foxo1 inhibition. We also examined the expression level of vegfaa after AS1842856 treatment; the results suggested that foxo1 inhibition did not affect the expression of vegfaa.

**Author response image 1. sa4fig1:** 

(2) In this manuscript, the authors performed single endothelial cell sequencing in glucose-treated embryos, and found reduced foxo1a expression and upregulated marcksl1a . Based on these data, the authors demonstrated that glucose and NMS-induced excessive angiogenesis through the foxo1a-marcksl1a pathway. The authors must conduct endothelial scRNA-seq in NMS-treated embryos, and analyze and compare the datasets with scRNA-seq datasets from glucose-treated endothelial cells, considering the focus of the paper. In addition, ASBs have been suggested as healthy alternatives to sugar-sweetened beverages. The authors also need to examine carefully whether metabolic gene programs are altered in glucose-treated endothelial cells, which was mentioned in Jorgens et al paper.

Thank you very much for your constructive suggestions. We have performed the whole embryo transcriptome sequencing after high D-glucose and L-glucose treatment. We analyzed and compared the differentially expressed genes of control, high D-glucose-treated, and high L-glucose-treated embryos. The results revealed that 1259 and 1074 genes were up-regulated significantly in high D-glucose and high L-glucose treated embryos, respectively, compared with control.

We also analyzed some metabolic-related genes and found that some genes involved in gluconeogenesis, glycolysis, and oxidative phosphorylation were significantly changed. The results were integrated into supplementary Figure12 and 13 of the revised manuscript.

(3) Glucose or NMS treatments induce the hyperbranched endothelial vessels from the dorsal aorta and ISVs but not cardinal veins. In Figure 4i, the arterial and capillary cell population is increased in glucose-treated embryos, but the venous cell population seems to be reduced. The authors need to check whether arterial/venous differentiation and proliferation are affected in glucose- and NMS-treated embryos.

Thank you for your suggestions. We examined arterial/venous differentiation based on Tg(*flt1BAC:YFP::kdrl:ras-mCherry*) zebrafish line, in which the YFP is mainly expressed in arterial Endothelial cells. We found the endothelial cells of excessively formed blood vessels induced by high glucose treatment are mainly arterial (Figure 2j). This might explain why the arterial and capillary cell population was increased in glucose-treated embryos.

(4) The manuscript proposes that excessively branched vessels within ISVs arise from the ectopic activation of quiescent endothelial cells (ECs) into tip cells. To confirm this process, the authors need to detect some specific tip cell markers to demonstrate their ectopic activation.

Thank you for your constructive suggestions. We have performed the analysis of single-cell RNA-seq data, and the results showed that the tip cell marker genes such as *esm1*, *apln*, and *cxcr4a* were significantly up-regulated in arterial and capillary ECs after high glucose treatment. The results were integrated into Figure 3 of the revised manuscript.

(5) Disaccharides such as lactose, maltose, and sucrose did not exhibit a notable induction of excessive angiogenic phenotype. However, the specific treatment concentrations utilized in the study were not delineated. Therefore, further investigation is warranted to determine whether increased disaccharide concentrations can cause vascular hyperbranching phenotype.

Thank you very much for the suggestions. We’ve described the concentrations of monosaccharides and disaccharides in the materials and methods section of the revised manuscript. Following the suggestion, we treated zebrafish embryos with a higher concentration of the disaccharide. The results showed that higher concentrations of disaccharide treatment also caused excessive angiogenesis in zebrafish embryos. These results were integrated into supplementary Figure 8 of the revised manuscript.

(6) The authors claim that glucose and NMS (such as L-glucose) induce excessive angiogenesis through the foxo1a-marcksl1a pathway. Following exposure to elevated glucose levels, a substantial down-regulation of foxo1a was observed in arterial and capillary endothelial cells. This down-regulation led to the release of foxo1a inhibition on marccksl1a, subsequently resulting in an augmented expression of marccksl1a and the manifestation of a vascular phenotype. Consequently, it is imperative to investigate whether the foxo1a overexpression can attenuate marccksl1a expression and mitigate the vascular phenotype induced by monosaccharides. Sufficient data support is needed for the conclusion that monosaccharides induce angiogenesis via the foxo1a-marcksl1a pathway.

Thank you very much for your constructive suggestions.

We confirmed the expression of *marcksl1a* in foxo1a-overexpressed embryos. The results indicated that foxo1a overexpression significantly attenuated *marcksl1a* expression. The results were integrated into Figure 8c. We also performed the rescue experiments, which indicated that overexpression of *foxo1a* partially mitigated the excessive angiogenesis induced by high glucose treatment. These results were integrated into Figure 6 of the revised manuscript.

Minor corrections:(1) Figure 2i, j has no corresponding graphs.

We’ve made the change in Figure 2.

(2) Figure 2h has no vertical coordinates.

We’ve made the change in Figure 2.

(3) All Figures should be referenced within the manuscript.

We’ve checked our manuscript carefully and made the corrections.

(4) The concentrations of monosaccharides and disaccharides employed in this study must be distinctly elucidated within the manuscript and annotated using the internationally recognized unit notation.

We’ve checked our manuscript carefully and described the concentrations of monosaccharides and disaccharides in the revised materials and methods section.

**Reviewer #3:**
(1) A possible limitation of the study is that the mechanism leading to angiogenesis in the retinal circulation and in peripheral vasculature is certainly different as diabetes is associated with excessive angiogenesis in the retina and a defect in angiogenesis in the peripheral circulation as shown by a reduced post-ischemic revascularization (see Silvestre et al.: DOI: 10.1152/physrev.00006.2013).

Thank you very much for your suggestions. As you said, the peripheral blood vessel model in this study does not fully represent individuals with diabetic retinopathy, which is a limitation. However, from a specific view, the phenotype and mechanism of excessive angiogenesis of peripheral blood vessels in the high glucose model may provide a reference for excessive angiogenesis in the retina; they might have similar etiology and regulation mechanisms in excessive angiogenesis.

(2) Another limitation is that angiogenesis in the embryo is not fully representative of the excessive angiogenesis observed in the diabetic retinal circulation. It would be of interest to analyse the retinal vascular tree in adult fish submitted to high glucose and to ASB.

In our future study, we will try to observe the angiogenesis phenotype in the diabetic retina and improve the disease model.

(3) Line 52: "Endothelial cell dysfunction (ECD)" instead of "Endothelial dysfunction (ECD)".

We’ve made the correction in the revised manuscript.

(4) The authors should elaborate more on the observation showing that L-glucose, D-mannose, D-ribose, and L-arabinose, which could not be digested by animals, also induce excessive angiogenesis. Is the effect indirect?

In the current manuscript, we conducted an in vivo live imaging analysis to show the phenotype of excessive angiogenesis caused by those noncaloric monosaccharides. However, we did not find differences in the phenotypes of embryos treated with noncaloric and caloric monosaccharides. Therefore, we supposed that the mechanisms underlying the phenotypes were similar. The effect might be indirect.